# Posteromedial thalamic nucleus activity significantly contributes to perceptual discrimination

**Jia Qi**, **Changquan Ye, Shovan Naskar, Ana R. Inácio, Soohyun Lee** *

Unit on Functional Neural Circuits, National Institute of Mental Health, National Institutes of Health, Bethesda, Maryland, United States of America

* soohyun.lee@nih.gov

**Data Availability Statement:** All relevant data is available within the manuscript and Supporting Information files.

**Funding:** All authors (SL, JQ, CY, SN, ARI) were supported by the Intramural Research Program of

## Abstract

Higher-order sensory thalamic nuclei are densely connected with multiple cortical and subcortical areas, yet the role of these nuclei remains elusive. The posteromedial thalamic nucleus (POm), the higher-order thalamic nucleus in the rodent somatosensory system, is an anatomical hub broadly connected with multiple sensory and motor brain areas yet weakly responds to passive sensory stimulation and whisker movements. To understand the role of POm in sensory perception, we developed a self-initiated, two-alternative forced-choice task in freely moving mice during active sensing. Using optogenetic and chemogenetic manipulation, we show that POm plays a significant role in sensory perception and the projection from the primary somatosensory cortex to POm is critical for the contribution of POm in sensory perception during active sensing.

## Introduction

Higher-order sensory thalamic nuclei are situated at the interface of cortical and subcortical areas. The intricate anatomical interconnections suggest the significance of higher-order thalamic nuclei in the contextual modulation of sensory information [1–7]. However, the behavioral relevance of higher-order thalamic nuclei in sensory perception remains unclear.

The posteromedial thalamic nucleus (POm), a higher-order thalamic nucleus in the somatosensory system, is considered one of the key nodes in integrating sensory-motor processing [4,8] and is involved in sensory learning-related synaptic plasticity in the cortex [9–12]. POm receives "bottom-up" sensory inputs from the brainstem [13–16] and "top-down" cortical inputs from multiple cortical areas, including somatosensory and motor cortices [13,17–19], in addition to the projections from the superior colliculus [20,21] and zona incerta [22–24]. Despite these diverse inputs, POm neurons respond weakly to passive sensory stimulation, likely because of the high susceptibility of POm neurons to brain states [3,7,25]. Under anesthesia, for instance, the spontaneous and sensory-evoked activity of POm neurons are markedly reduced [26]. The robust anatomical interactions with sensory and motor cortices suggested that POm may encode self-generated whisker movements. However, the postulated role of POm in encoding whisking behavior has been challenged with studies demonstrating

National Institute of Mental Health, National Institutes of Health (ZIAMH002959). The funders had no role in study design, data collection and analysis, decision to publish, or preparation of the manuscript.

**Competing interests:** The authors have declared that no competing interests exist.

**Abbreviations:** ACSF, artificial cerebrospinal fluid; AP, action potential; CNO, clozapine-N-oxide; DREADD, designer receptors exclusively activated by designer drugs; i.p., intraperitoneally; LP, lateral posterior; PFA, paraformaldehyde; POm, Posteromedial thalamic nucleus; RMP, resting membrane potential; RT, room temperature; TDI, texture difference index; VPm, ventral posteromedial nucleus; WR, water reward; 2AFC, two-alternative forced choice.

that POm activity is weakly modulated by voluntary whisking behavior, although artificial whisking induced by electrical stimulation to the facial nerve of anesthetized animals evokes stronger activity in POm during whisking [3,27–30]. The dense anatomical interactions in the face of relatively weak sensory responses and poor ability to encode whisking behavior raises an important question regarding the role that POm plays in sensory perception during active sensation. To address this question, we have developed a whisker-dependent, self-initiated, two-alternative forced choice (2AFC) task in freely moving animals. Using this simple yet robust texture-dependent sensory perception task combined with the manipulation of POm activity and its input sources, we show that POm significantly contributes to sensory discrimination and the projections from the primary somatosensory cortex to POm is critical for the role of POm in perceptual sensory discrimination during active sensing.

## Results

### Whisker-dependent texture discrimination in freely moving mice

To investigate the role of POm in sensory perception during natural behavioral conditions, we designed a whisker-dependent texture discrimination task in the form of a self-initiated, 2AFC task in freely moving mice (**Fig 1A** and **1B**). Animals initiated each trial by poking their nose into the center port of a task chamber. Following trial initiation, two different textures were simultaneously presented for 1 s. Animals were trained to sense two different textures with their whiskers and to choose one of the two side ports for a reward that was associated with a target texture. There was no punishment for incorrect choices or time-out before the animals initiated a subsequent trial. The animal's behavior was monitored and video recorded during the task. Sandpapers with four different grits (P120, P180, P280, and P400) were used for the texture discrimination task (**Fig 1B**). The four sandpapers with different grits were mounted on a cube, and two cubes presented two different textures for each trial. Initially, animals were trained to discriminate two different textures, P120 and P400. Once the animals reached performance accuracy of 70% on three consecutive sessions, the animals were considered to have learned the task (**Fig 1C**). Then, the target texture (P120 or P400) was presented with one of the other three different textures in a pseudorandomized order (**Fig 1B**). The target texture was counterbalanced across mice. To describe the different levels of the task difficulty, we employed a texture difference index (TDI; TDI = (average particle size of texture1 − average particle size of texture 2) / 100). Task performance increased as the TDI increased (**Fig 1D**). To avoid the influence of visual cues, the behavioral task was conducted in a dark environment. To further confirm that the animals use their whiskers to perform the task, we bilaterally trimmed all the whiskers at the end of all experiments. After trimming, the animals performed the task at chance level, indicating that the animals rely on their whiskers to perform the task (**Fig 1E**; before: $82.15 \pm 1.10\%$ versus after: $52.97 \pm 0.74\%$; t = 22.39, df = 27, $n$ = 28, $P < 0.0001$). These results demonstrate that our self-initiated texture discrimination task is a simple yet robust behavior task to study whisker-dependent sensory perception in freely moving animals.

### A role of POm in perceptual sensory discrimination

POm neurons show slower and weaker responses to whisker stimulation, while ventral posteromedial nucleus (VPm) neurons, a first-order thalamic nucleus of the somatosensory system, strongly respond to passive whisker stimulation with high spatiotemporal fidelity [26,31,32]. Considering that POm receives converging inputs from diverse brain areas, weak responsiveness of POm neurons raises questions on the role of POm in sensory perception, and behavioral conditions and stimuli that engage POm neurons. However, most studies were

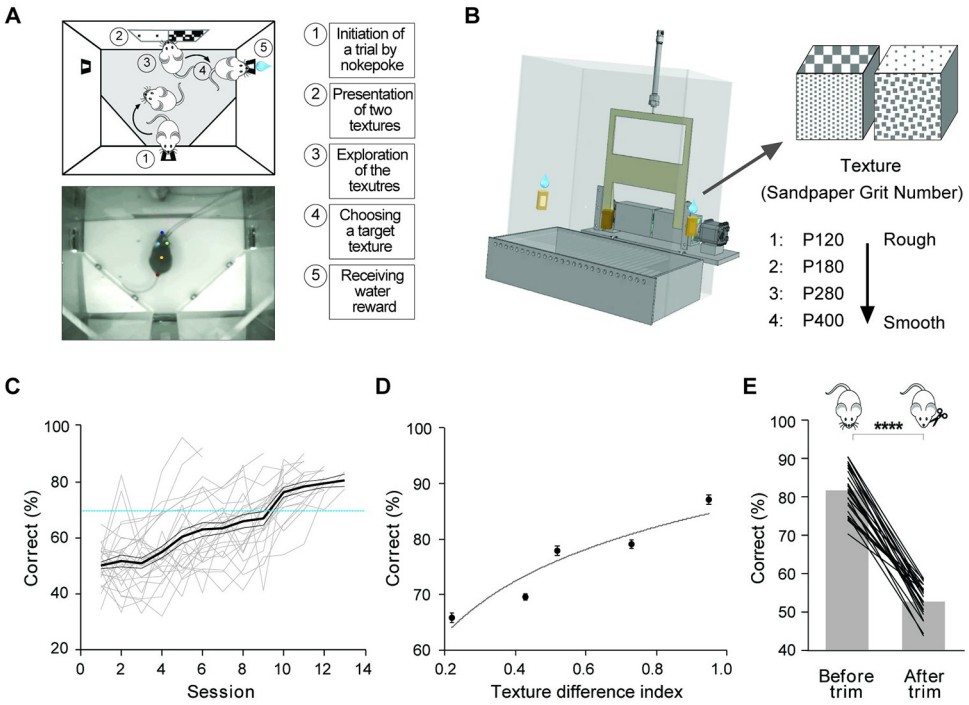

**Fig 1. Self-initiated, whisker-dependent texture discrimination task in freely moving mice. (A)** Schematic of texture discrimination task in freely moving mice. Upper, top view cartoon demonstrates the sequence of the self-initiated whisker-dependent texture discrimination 2AFC task. Lower, top view image of a mouse with the optical fibers approaching sensory zone. The color dots are post hoc labeled by DeepLabCut tracking software to indicate the mouse's nose, ears, back, and tail. Left, task sequence. **(B)** Schematic of texture presentation. Left, behavioral chamber design with nose poking sensors, retractable window, two rotatable cubes with four different textures on four surfaces of each cube, and water reward ports. Right, four different grits of sandpapers (P120, P180, P280, and P400) are mounted on each cube. Two cubes present two different textures in each trial. **(C)** Learning curve for the mice trained on texture discrimination task (texture P120 vs. P400). Gray line represents a learning curve for individual mouse (*n* = 28) over multiple sessions. The thick black line represents average performance across mice, and the thin black line indicates SEM. Blue dotted line indicates 70% of correct trials in task performance. **(D)** Dependence of task performance on texture difference. Texture difference index (TDI) is calculated as the difference of two texture grits divided by 100. Task performance increases as TDI increases. **(E)** Behavioral performance decreases to chance level after bilateral whisker trimming ($P < 0.0001$, *n* = 28, paired *t* test). Gray bars indicate average across 28 mice, and black lines represent an individual mouse. The data in Fig 1C–1E, can be found in S1 Data.

conducted using passive sensory stimulation under anesthetized or head-fixed conditions, thus making it difficult to understand the function of POm during active sensing in awake animals. To address this question, we suppressed POm activity with a chemogenetic method using designer receptors exclusively activated by designer drugs (DREADDs) while the freely moving animals were performing the self-initiated texture discrimination task (**Figs 2** and **S1**). We first tested the efficacy of clozapine-N-oxide (CNO) on hM4D(Gi)-expressing POm neurons. We confirmed that CNO application significantly lowered the resting membrane potential and increased the rheobase of hM4D(Gi)-expressing POm neurons (**S2 Fig**, resting membrane potential, control −67.93 ± 1.04 Vm versus CNO −70.95 ± 1.34 Vm, $P = 0.0059$; rheobase, control 33.00 ± 7.53 pA versus CNO 69.36 ± 13.28 pA, $P = 0.0020$, *n* = 11 cells from 4 mice). We bilaterally injected inhibitory DREADDs (AAV-hSyn-hM4D(Gi)-mCherry) [33] into POm three weeks prior to the behavioral task. We then interleaved the behavioral task sessions with CNO (3 mg/kg, intraperitoneally (i.p.)) or saline injection (**Fig 2A**). The direct suppression of POm neurons with chemogenetic approach significantly impaired the animal's performance in the texture discrimination task (**Fig 2B**; saline 83.09 ± 1.90% versus CNO

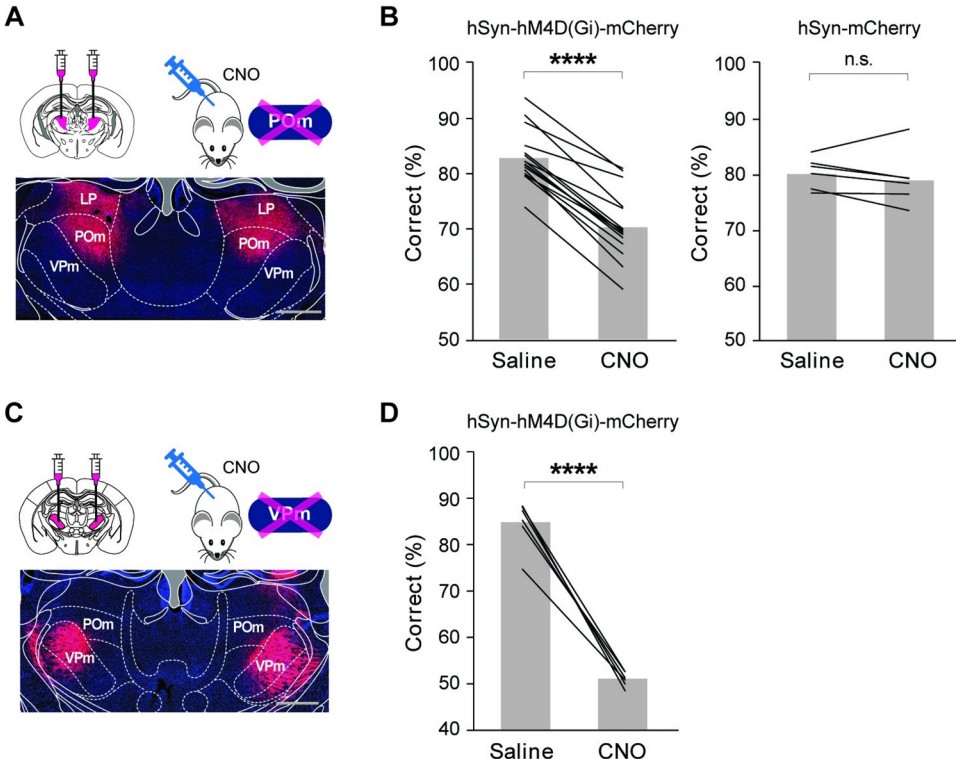

**Fig 2. Role of POm in sensory discrimination during active sensing. (A)** Schematic of experimental design and a representative image showing the expression of hM4D(Gi)-mCherry in POm. Scale bar = 500 μm. **(B)** Chemogenetic suppression of POm significantly impairs the task performance of texture discrimination task. CNO injection, but not saline injection, significantly attenuates the behavior performance of the hM4D(Gi)-mCherry group mice in the whisker-dependent 2AFC task ($P < 0.0001$, $n = 16$, paired $t$ test). In the AAV-hSyn-mCherry control group mice, CNO injection shows no effect on task performance ($P = 0.38$, $n = 6$, paired $t$ test). **(C)** Schematic of experimental design and a representative image showing the expression of hM4D(Gi)-mCherry in VPm. **(D)** Chemogenetic suppression of VPm results in chance level performance. ($P < 0.0001$, $n = 6$, paired $t$ test). Scale bar = 500 μm. The data in Fig 2B and 2E can be found in S1 Data. CNO, clozapine-N-oxide; POm, posteromedial thalamic nucleus; VPm, ventral posteromedial nucleus; 2AFC, two-alternative forced choice.

$70.55 \pm 2.38\%$, t = 16.15, df = 15, $P < 0.0001$, $n = 16$, paired $t$ test). To test the off-target effect of CNO, we also injected a control virus (AAV-hSyn-mCherry) into POm in a different set of animals and administered CNO or saline. Task performance between saline and CNO-injected groups did not differ (**Fig 2B**; saline $80.14 \pm 1.04\%$ versus CNO $79.03 \pm 1.83\%$, t = 0.96, df = 5, $P = 0.38$, $n = 6$, paired $t$ test). While the suppression of POm activity significantly lowered the rate of correct response, the animals' performance was still higher than chance levels. This is because VPm was still intact. To test the role of VPm in sensory perception tasks, we bilaterally injected inhibitory DREADDs into VPm (**Fig 2C**). As expected, chemogenetic suppression of VPm resulted in chance level performance (**Fig 2D**, $P < 0.0001$, $n = 6$, paired $t$ test). However, the significant role of POm in sensory discrimination is unexpected given the slow and weak responsiveness of POm neurons to passive sensory stimulation and the low spatial resolution of POm receptive fields [26,31,32]. To ensure that the expression of hM4D(Gi) receptors was restricted within POm and did not spread to VPm, we validated the expression of hM4D(Gi)-mCherry in POm and VPm from each mouse (**S3 Fig**). While our injection was mostly restricted within POm and reliably avoided VPm, we consistently detected the expression of hM4D(Gi)-mCherry in the lateral posterior (LP) thalamic nucleus that is situated just above POm (**Fig 2A**). LP, the higher-order thalamic nucleus in visual system of rodents, is highly

sensitive to diverse behavioral contexts [5,6,34–36]. To address whether the expression of hM4D(Gi) receptors in LP affect the animals' performance during the texture discrimination task, we bilaterally expressed hM4D(Gi) receptors only in LP but not in POm (**S4 Fig**). Suppression of LP activity with CNO injection did not affect the task performance, demonstrating that the impairment in the animals' performance in the texture discrimination task is due to the suppression of POm activity and not because of the alteration of LP activity (**S4 Fig**; saline 81.40 ± 1.37% versus CNO 80.30 ± 1.26%; t = 1.32, df = 5, $P$ = 0.24; $n$ = 6, paired $t$ test). In summary, our results demonstrate that POm, a higher-order thalamic nucleus in the somatosensory system, plays a significant role in texture discrimination during active sensing.

## Corticothalamic inputs from S1 are necessary for the role of POm in perceptual sensory discrimination

We next asked which input sources are critical for the contribution of POm in perceptual sensory discrimination. POm receives not only afferents originating from the brainstem, mostly from spinal trigeminal interpolaris nucleus (Sp5i), through the paralemniscal pathway [13,14] but also inputs from multiple cortical areas, including motor and somatosensory cortices via the descending pathway [13,17–19,37]. To identify the main excitatory input source that enables POm to play a significant role in sensory perception, we manipulated the afferents to POm with an optogenetic approach. We asked whether the suppression of the axon terminals from the major afferents, Sp5i, M1/M2, and S1 to POm affect the sensory perception of the animals during the texture discrimination task (**Fig 3**).

To target corticothalamic inputs from S1 and M1/M2 to POm, we employed an Rbp4-cre mouse line in which Cre is expressed in layer 5 (L5) pyramidal neurons [38], the main cortical layer that projects to POm [37]. We bilaterally injected AAV vectors expressing halorhodopsin (AAV5-EF1a-DIO-eNpHR3.0-eYFP) [39] into motor cortex (M1/M2), and primary somatosensory cortex (S1) in separate animals (**Fig 3C** and **3E**). To minimize mouse strain-dependent effects on mouse behavior, we used the Rbp4-cre mouse line in targeting Sp5i neurons despite constitutive AAV vectors (AAV5-hSyn-eNpHR3.0-eYFP) were injected to Sp5i (**Fig 3A**). We also bilaterally expressed hM4D(Gi) receptors in POm of these three groups of mice. This strategy allowed us to independently control the activity of POm neurons chemogenetically and the excitatory inputs to POm optogenetically. Optical fibers (200 μm) were bilaterally implanted above POm to deliver 590 nm light (8 to 10 mW) for optogenetic suppression of the axon terminals from the main afferents (**Fig 3A, 3C**, and **3E**). The light application was triggered by trial initiation and terminated once an animal made a choice by poking its nose into the water port. To confirm the optogenetic suppression of axon terminals innervating POm, we performed in vivo extracellular recording from POm using an optrode (optic fiber 100 μm) (**S5 Fig**). We recorded spontaneous activity of POm neurons, while applying optogenetic stimulation to POm innervating M1/M2 terminals in awake, head-fixed mice. Optogenetic application significantly suppressed spontaneous spiking activity (21 out of 109 POm units, baseline 7.49 ± 1.25 Hz versus 590 nm ON 4.99 ± 0.95 Hz, $P$ < 0.001, paired $t$ test).

Bilateral optogenetic suppression of Sp5i axon terminals in POm showed no effect on the rate of correct response (**Fig 3B**; LED off 80.36 ± 0.59% versus LED on 79.50 ± 0.51%, t = 1.456, df = 4, $P$ = 0.22, $n$ = 5, paired $t$ test), while chemogenetic suppression of POm neurons with CNO injection in the same animals did significantly lower the task performance (**Fig 3B**; saline 82.25 ± 0.60% versus CNO 68.82 ± 0.42%, t = 44.04, df = 4, $P$ < 0.0001, $n$ = 5, paired $t$ test). Expression of halorhodopsin was localized in the Sp5i and was not detected in the principal trigeminal nucleus (Pr5), which is the main brainstem nucleus for lemniscal

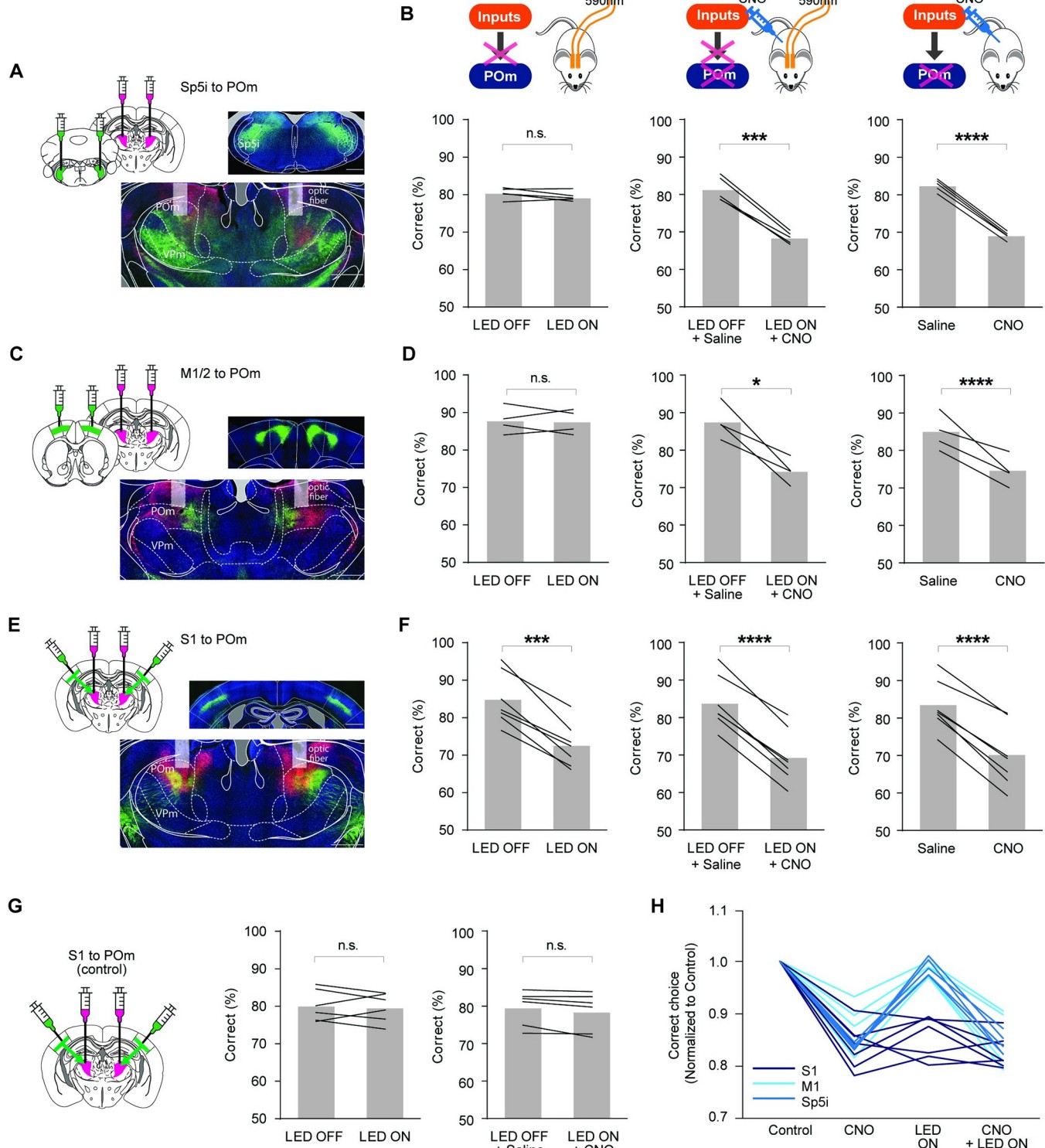

**Fig 3. Corticothalamic inputs from S1 are necessary for the role of POm in perceptual sensory discrimination. (A, B, C, E)** Experimental strategy with chemogenetic and optogenetic manipulation of POm and inputs to POm. Left, schematic of experimental design. Right, representative images showing the expression of hM4D(Gi)-mCherry in POm (A, C, and E) and eNpHR3.0-eYFP in Sp5i (A), M1/M2 (C), and S1 (E). In three different groups of Rbp4-cre mice, Cre-dependent eNpHR-eYFP was injected into L5 of S1 or M1/M2, while AAV-hSyn-eNpHR- eYFP was delivered into Sp5i. In all three groups, AAV-hSyn-hM4D(Gi)-mCherry was injected into POm. All of the viral injections were bilateral, and the optical probes were bilaterally implanted above POm. **(B)** Role of Sp5i inputs to POm in sensory discrimination. Left, optogenetic suppression of Sp5i axon terminals in POm does not affect task performance (*P* = 0.22, *n* = 5,

paired *t* test), while chemogenetic inhibition of POm (right) or the combined inhibition (middle) significantly reduce task performance (*P* = 0.0002, *n* = 5, paired *t* test). **(D)** Role of M1/M2 inputs to POm in sensory discrimination. Left, optogenetic suppression of M1/M2 axon terminals in POm does not affect task performance (*P* = 0.84, *n* = 4, paired *t* test), while chemogenetic suppression of POm (right) or the combined suppression (middle) significantly reduce task performance (*P* = 0.0002, *n* = 4, paired *t* test). **(F)** Role of S1 inputs to POm in sensory discrimination. Upper, schematic of the optogenetic (left), chemogenetic (right), or the combined (middle) manipulation strategies. Left, optogenetic suppression of S1 axon terminals in POm significantly attenuates task performance (*P* = 0.0002, *n* = 7, paired *t* test). Decrease in task performance during chemogenetic suppression of POm (right, *P* < 0.0001, *n* = 7, paired *t* test) or combined both optogenetic and chemogenetic suppression (middle, *P* < 0.0001, *n* = 7, paired *t* test) is comparable to the level of the attenuation during optogenetic suppression of S1 axon terminals innervating POm. The gray bar indicates the average across mice, and the black line represents an individual mouse. **(G)** AAV5-EF1a-DIO-eYFP were bilaterally injected into S1 and AAV5-CAG-mCherry into POm of control group (Rbp4-cre). Neither optogenetic nor combined suppression had effects on task performance in the control group (left panel, *P* = 0.6629; right panel, *P* = 0.0695; *n* = 6, paired *t* test). **(H)** Suppression of corticothalamic inputs from S1 to POm significantly lowers sensory discrimination. Normalized correct choice of three groups in three different manipulations. While chemogenetic suppression of POm significantly reduces correct choice in all three groups, only optogenetic suppression of S1 inputs to POm affects task performance similar to POm suppression, but not the M1/M2 and Sp5i inputs to POm groups. Scale bar = 500 μm in (A, C, and E). The data in Fig 3B, 3D, 3F, 3G, and 3H can be found in S1 Data.

pathway. Consistent with previous studies [16,40,41], we detected the innervation of Sp5i both in POm and in VPm (**Fig 3A**).

Similar to Sp5i inputs to POm, suppression of the axon terminals from M1/M2 did not change the animals' performance compared to the light-off conditions (**Fig 3D**; LED off 87.96 ± 1.54% versus LED on 87.67 ± 1.41%, t = 0.2139, df = 3, *P* = 0.84, *n* = 4, paired *t* test). However, CNO-induced suppression of POm activity in the same animals significantly lowered task performance (**Fig 3D**; saline 84.45 ± 1.82% versus CNO 74.25 ± 1.51%, t = 4.387, df = 3, *P* = 0.022, *n* = 4, paired *t* test). This result demonstrates that inputs from Sp5i and M1/M2 do not transmit perceptual information to POm. We wondered whether cre-dependent expression of NpHR within a Rbp4-cre mouse line may limit the expression of NpHR only within a subpopulation of M1/M2 L5/6 cells projecting to POm and, thus, result in the lack of optogenetic suppression effect in task performance. We first examined the laminar distribution of POm-projecting neurons by injecting AAVretro-CAG-Cre to POm of an Ai14 mouse (**S6 Fig**). While we detected POm-projecting neurons from both L5 and L6, we found more abundant POm-projecting cells in L5 (**S6 Fig**). To further test the role of M1/M2 inputs to POm in texture discrimination, we constitutively expressed NpHR from neurons in M1/M2 of wild-type mice. This allows more broad and unbiased expression of NpHR in M1/M2 cells. Optogenetic suppression of these inputs to POm resulted in no detectable impact on task performance, suggesting that M1/M2 inputs to POm do not play an important role in sensory discrimination (**S6 Fig**; *P* = 0.43, *n* = 4, paired *t* test).

In contrast to Sp5i and M1/M2 inputs, suppression of cortical inputs from S1 to POm (**Fig 3E**) significantly reduced task performance (**Fig 3F**; LED off 84.89 ± 2.46% versus LED on 72.63 ± 2.05%, t = 8.272, df = 6, *P* = 0.0002, *n* = 7, paired *t* test). The reduction in the rate of correct responses during optogenetic perturbation of S1 inputs to POm was similar to the decrease when the activity of POm was suppressed with CNO in the same animals (**Fig 3F**; saline 82.93 ± 2.30% versus CNO 69.65 ± 2.92%, t = 12.3, df = 6, *P* < 0.0001, *n* = 7, paired *t* test). To further test whether S1 inputs are the main source in conveying the perceptual information during the texture discrimination task, we suppressed POm activity with CNO injection, in addition to the optogenetic perturbation of S1 inputs. The combined perturbation did not further reduce performance (**Fig 3F**; LED off + saline 83.93 ± 2.47% versus LED on + CNO 69.47 ± 2.51%, t = 14.61, df = 6, *P* < 0.0001, *n* = 7, paired *t* test). Moreover, no changes in behavioral performance were found in the control group (**Fig 3G**; LED off 80.21 ± 1.57% versus LED on 79.67 ± 1.45%, t = 0.4629, df = 5, *P* = 0.66, *n* = 6, paired *t* test; LED off + saline 79.65 ± 1.73% versus LED on + CNO 78.52 ± 1.93%, t = 2.303, df = 5, *P* = 0.0695, *n* = 6, paired *t* test). The normalized behavioral performance (**Fig 3H**) shows that the role of POm in sensory discrimination is mainly inherited from primary sensory cortical projections but not

from motor cortical projections or brainstem ascending pathway. Together, our finding suggests that the cortical inputs from S1 are mostly responsible for the role of POm in perceptual sensory discrimination.

To examine whether CNO and optogenetic suppression alters the animals' motivation and movement, and consequent impairment in task performance, we analyzed behavioral parameters of the sensory discrimination task. We analyzed total trial numbers in each session, trial duration, intertrial interval, movement velocity, and duration of active sensing against the presented textures in each trial using DeepLabCut [42]. These parameters under optogenetic and chemogenetic manipulation conditions were comparable to those under control conditions (S7 Fig). The results suggest that the impaired performance in the discrimination task was not due to movement deficit or the animals' motivation.

## POm is more strongly engaged as the level of task difficulty increases

The performance of the texture discrimination task strongly depends on the level of the task difficulty (Fig 1D). We asked whether POm suppression differentially affects animals' performance as the level of task difficulty changes. We separated trials into low (texture 1 versus 4), medium (textures 1 versus 3 or 4 versus 2), and high (textures 1 versus 2 or 4 versus 3) levels of task difficulty based on the similarity of the two presented textures. While the rate of correct responses decreased in all three levels during CNO sessions, the rate of error responses disproportionately increased in medium and high task difficulty trials compared to changes in low difficulty trials (Fig 4A; high 16.97 ± 2.04% versus low 8.03 ± 1.32%, t = 2.899, df = 6, $P$ = 0.0274; medium 15.78 ± 1.98% versus low 8.03 ± 1.32%, t = 2.584, df = 6, $P$ = 0.0415, $n$ = 7, paired $t$ test). Similar to POm suppression, the changes in the rate of incorrect responses during trials of medium and high task level of difficulty were significantly greater than the low level of difficulty trials when S1 input to POm was suppressed (Fig 4B; high 15.20 ± 2.31% versus low 6.04 ± 1.83%, t = 2.866, df = 6, $P$ = 0.0286; medium 12.18 ± 1.55% versus low 6.04 ± 1.83%, t = 3.617, df = 6, $P$ = 0.0111, $n$ = 7, paired $t$ test). These findings support that the contribution of POm in sensory perception is more significant in higher demand of texture discrimination ability, and the S1 inputs to POm are critical in supporting the role of POm in higher demand of discriminability.

## Discussion

Diverse functions of the higher-order thalamic nuclei have been suggested, largely based on their dense interactions with multiple cortical and subcortical areas in the brain. Yet, whether and how the higher-order thalamic nucleus is engaged in perceptual decision during active sensing has not been addressed. Here, we investigated the role of a higher-order thalamic nucleus in perceptual discrimination during active sensation in the vibrissa-related somatosensory system. We report that suppression of POm activity significantly affects behavioral performance in a whisker-dependent sensory discrimination task, leading to a progressively large impairment as the difficulty of the task increased. We further demonstrate that the projection from the primary somatosensory cortex is required for the contribution of POm to perceptual sensory discrimination.

Our finding of the significance of POm activity in sensory discrimination is rather surprising since whisker-driven POm activity is relatively poor in spatial (large receptive field) and temporal (long latency) resolution, as well as in response magnitude compared to the response of VPm, a first-order thalamic nucleus in the somatosensory system [32,43]. We limited the viral injection of hM4D(Gi)-mCherry to POm to avoid the contamination of the viral injection to VPm. Therefore, our results may underestimate the contribution of POm in sensory discrimination. However, previous studies were mostly performed under anesthesia, passive

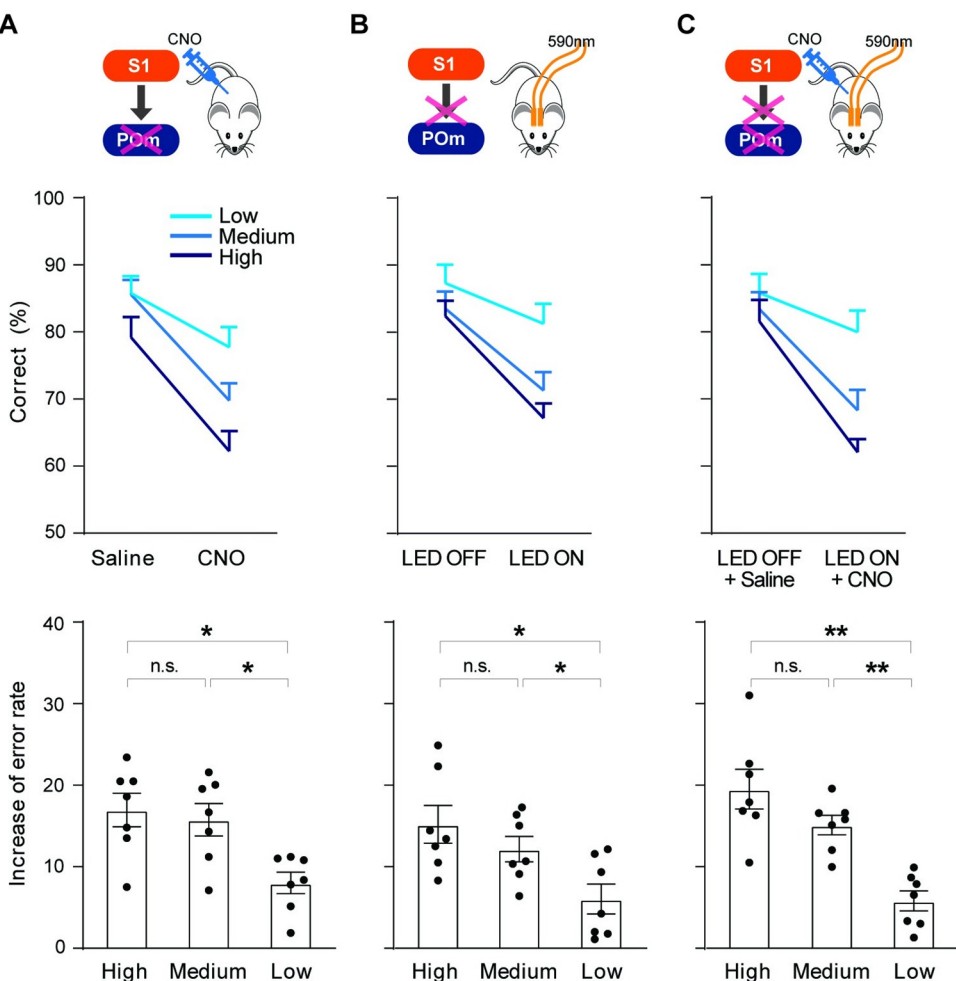

**Fig 4. Involvement of POm increases as discriminability of texture stimuli decreases. (A)** Effect of POm suppression on the different levels of task difficulty in the texture discrimination task. Upper, schematic of experiment strategy. Middle, POm suppression differentially affects task performance based on the level of task difficulty. Lower, increase of the error rate is significantly higher in high and medium task difficulty trials (high vs. low, $P = 0.0274$; medium vs. low, $P = 0.0415$, $n = 7$, paired $t$ test). Error rate is calculated by subtracting the percentage of correct responses from 100. An increase in error rate is calculated by subtracting the error rate of the correlated control group from the error rate of the manipulation group. **(B)** Suppression of S1 axons in POm differently affect task performance based on the level of task difficulty (high vs. low, $P = 0.0286$; medium vs. low, $P = 0.0111$, $n = 7$, paired $t$ test). Plotting conventions are same as (A). **(C)** Effect of combined suppression of POm and S1 axons in POm does not differ from POm suppression or suppression of S1 inputs to POm (high vs. low, $P = 0.0024$; medium vs. low, $P = 0.0018$, $n = 7$, paired $t$ test). Plotting conventions are same as (A). The data in Fig 4A–4C can be found in S1 Data.

sensory stimulation, or an unengaged condition. Increasing evidence supports that the activity of POm is strongly modulated by behavior and brain states [3,7,25]. Our study implies that POm is more robustly engaged during goal-directed, active sensing under freely moving condition. Intriguingly, sensory response of VPm is dynamically modulated by voluntary movements via corticothalamic projection from L6 to VPm [44]. While alteration of POm activity could strongly affect sensory responses in the cortex [45,46] via its distinct connectivity to S1 [47–49], the activity of POm neurons during actively engaged behavioral conditions has not been well addressed. Future studies probing the activities of POm neurons and axon terminals from multiple brain areas during different behavioral contexts will provide deeper insight into how information flow within POm circuitry supports sensory perception.

POm responses to sensory stimulation are strongly regulated by the corticothalamic projections from L5 of S1, as suggested by the finding that L5 cortical inputs are the "driver" inputs to POm [13,17,18,50–52]. Our results demonstrate that L5 inputs from S1, but not L5 and L6 inputs from M1/M2 nor Sp5i input from bigeminal brainstem, enables the role of POm in perceptual decision-making. In agreement with our results, a recent fMRI study using optogenetic manipulation demonstrated that sensory-evoked activity in POm is mostly driven by S1, while inputs from M1 and S2 have almost no effect on the evoked sensory response in POm [53]. Furthermore, the fMRI study demonstrated that forelimb stimulation did not elicit sensory response in POm when S1 was ontogenetically suppressed. One of the hypothesized roles of M1 inputs to POm is controlling whisker movements during active sensing. In the current study, the suppression of M1 axon terminals in POm did not affect the animals' performance in the sensory discrimination task. Even though the M1 inputs to POm is suppressed, the animals may rapidly adjust their motor planning and strategy. Thus, it is possible that the inactivation of M1 terminals in POm may lead to no changes in the sensory discrimination. Intriguingly, POm suppression more severely affects the animals' task performance in the higher level of task difficulty trials, suggesting that POm is more significantly engaged in higher perceptual demands during sensory discrimination. Similar task difficulty dependent effect was observed upon the suppression of S1 inputs to POm.

How does POm contribute to sensory perception? It is unlikely that S1 inputs are processed locally within POm since the local synaptic interactions among POm neurons are weak [54]. Instead, POm may integrate S1 inputs with other information from different sources and route the combined information to perceptual decision-relevant cortical and subcortical areas via *trans*-thalamic projections [5,50,51,55,56]. Supporting this idea, a recent study of visual system [5] demonstrates that direct projections from primary visual cortex to other cortical areas and indirect projection via LP, *trans*-thalamic pathway, from V1 to other cortical areas convey distinct visual information.

The current study focused on the function of POm in the context of sensory perception. The optogenetic and chemogenetic manipulations in our study were applied to POm as a whole and did not parse the substructures within POm. However, growing evidence supports the idea that the higher-order sensory thalamic nucleus is not a homogeneous structure but is rather comprised of heterogeneous subregions that each has its own distinct input and output connectivity [13,34,54,56]. Our study also showed that the axons from M1/M2 target the medial part of POm while S1 axon terminals occupy the lateral part of POm, suggesting potential anatomical segregation within POm (**Fig 3C** and **3E**) in addition to subregions along the anteroposterior axis of POm [13]. Related to the structural heterogeneous subregions, POm is likely involved in a broad range of different functions, such as sensorimotor integration, sensory learning, attention, and prediction. Understanding how different subregions of POm are engaged in performing distinct functions will provide a better understanding of the structural and functional complexity of POm.

## Material and methods

### Animals

Both male and female Rbp4-cre mice (GENSAT) were used in this behavioral study ($n$ = 28, age: 8 to 26 weeks when the behavioral experiment was performed). Mice were group housed under 12 h:12 h reverse light:dark cycle (light off at 8 AM), 22°C with ad libitum access to food. After surgery, mice were singly housed. Behavioral training and task were performed in the dark cycle between 9 AM and 6 PM. Before the behavioral study, mice had restricted access to water (1 ml/day). When the body weight stabilized at 85% of initial weight, mice daily

performed a behavioral session lasting an hour in which they received water (0.5 to 1.2 ml). After the surgery, mice had free access to water for at least five days before the start of water restriction. All procedures were in accordance with protocols approved by the NIMH/NIH Animal Care and Use Committee.

## Virus injection and optical fibers implantation

Rbp4-cre mice were anesthetized in a chamber containing isoflurane (5%) (Absolute Anesthesia, VetEquip) and maintained under anesthesia (1.4 to 1.6% isoflurane, mixed with 1 L per min of oxygen) for surgical procedures and placed into a stereotactic frame (Stoelting). Body temperature was maintained at 37°C with a heating pad during the entire surgical procedure, and ophthalmic ointment was applied periodically to maintain eye lubrication. The skull was thinned with a dental drill, and a small craniotomy (approximately 150 μm in diameter) was made. Virus was injected using a glass pipette (approximately 10 μm inner diameter) attached with a microprocessor-controlled nanoinjector (WPI). Virus was delivered at 20 nl/min, and the glass pipette was kept in place for approximately 5 min before slowly retracted. Meloxicam (2 mg/kg) was subcutaneously injected for analgesia and anti-inflammatory purposes.

Rbp4-cre mice were bilaterally injected with AAV5-hSyn-hM4D(Gi)-mCherry ($1.2 \times 10^{-13}$ GC/ml, Addgene) into POm (AP −2.0, ML ± 1.3, DV −3.2 mm from bregma; 50 nl) and AAV5-EF1a-DIO-eNpHR3.0-eYFP ($3.2 \times 10^{-12}$ GC/ml, UNC) into the S1 L5 (AP −0.9, −1.1, −1.3, ML ± 3.4 at 18°, DV −0.65 mm from bregma; 150 nl in total per hemisphere), or motor cortex L5 (AP +0.8, +1.0, ML ± 1.0, DV −0.80 mm; 100 nl in total per hemisphere), or AAV5-hSyn-eNpHR3.0-eYFP ($2.3 \times 10^{-12}$ GC/ml, Addgene) into the Sp5i (AP −6.6, ML ± 1.8, DV −3.5, −3.8 mm from pia; 200 nl in total per hemisphere). Rbp4-cre mice were bilaterally injected with AAV5-hSyn-hM4D(Gi)-mCherry into the LP (AP −2.0, ML ± 1.3, DV −2.8; 50 nl) or VPm (AP −1.7, ML ± 1.65, DV −3.4 mm). The coordinates are based on The Mouse Brain in Stereotaxic Coordinates by Franklin and Paxinos (third edition). For control groups, AAV5-CAG-mCherry ($1.0 \times 10^{-13}$ GC/ml, Addgene) and AAV5-EF1a-DIO-eYFP ($1.0 \times 10^{-13}$ GC/ml, Addgene) were injected into POm and S1 of Rbp4-cre mice, respectively. To identify the laminar distribution of POm-projecting M1/M2 neurons, we injected AAVretro-CAG-cre ($2.3 \times 10^{-12}$ GC/ml, NINDS/NIH virus core) into POm of Ai14 mouse line. Optical fibers (200 μm, Doric Lenses) for optogenetic suppression were bilaterally implanted above POm (AP −2.0, ML ± 1.3, DV −2.9) and secured with dental cement (C&B Metabond).

## Behavioral test apparatus

An acrylic box (30 L × 20 W × 30 H, in cm) with an open top served as the building block for the behavioral testing system. An 8020 frame was situated in front of the box to accommodate the valves and water reservoir attachment for the water delivery system. The entire chamber and frame were mounted on an optical breadboard (Thor Labs). The box was outfitted with three custom-designed nose poke sensors with optical sensors. Two nose poke sensors included a water reward delivery (10 μl/50 ms) system controlled by a solenoid operated pinch valve. The front of the box was equipped with a pull-up window (2 × 8 cm) that presented the mouse with various textures in the form of sandpapers in P120, P180, P280, and P400 grits (3M). The sandpapers adhered to four sides of two cubes (4 cm) that were rotated by a servo motor (Robotis). An optical sensor was installed near the window to detect the texture sampling duration. Behavioral experiments were performed using a data acquisition interface (National Instruments) and customized Labview software (National Instruments) for synchronization with Plexon infrared (IR) video system.

## Training paradigm of whisker-dependent, self-initiated 2AFC texture discrimination task

Mice performed one training session per day. The entire behavioral task was performed in a dark environment. The movements of mice were recorded by Plexon infrared video system (30 frames/s) (S1 Fig).

Phase 0: Mice were trained for 1 or 2 sessions to reliably trigger water reward (WR) by nose poking the water port on the side walls. Once mice received more than 60 WR in 30 min per session, mice proceeded to the next phase.

Phase 1: Mice learned the sequence of the task: (1) initiate a trial by nose poke a trial initiation sensor to trigger 500 Hz sound cue for 1 s; (2) move to the texture discrimination window area, i.e., sensory zone (the window stays close), and break the IR beam; and (3) receive WR from a water port, which triggers reinforcement sound (1,000 Hz for 1 s). The water delivery was pseudorandomly presented in either water port, and light cue above the water port was used to guide the mice to avoid the left or right preference in mice. Once mice received more than 45 WR in a 1-h session and the average rate of correct choice was above 85%, mice proceeded to the next phase.

Phase 2: Mice were trained to detect the textures using their whiskers and develop the association between the target texture and WR, an operant texture discrimination conditioning. Mice followed the same sequence of the task as in Phase 1. However, after mice initiated a trial, a window in the sensory zone was opened, and two different textures were simultaneously presented (Fig 1A and 1B). The window was positioned 2 cm above the floor to enforce the mice to use their whiskers to sense the two textures (P120 and P400) and not their forepaws. Either rough (P120) or smooth (P400) sandpaper served as a target stimulus, which was counterbalanced across the mice. Once the accumulated time of IR beam break in the sensory zone reached 1 s, the window was closed. Mice reported a target texture by nose poking a water port on the target texture side and received WR. Light cue for a correct choice was not available in this phase. Mice did not receive WR when it made an incorrect choice. There was no punishment for an incorrect choice. Target texture conditioning was established with a block design training, starting from 20 trials to 5 trials per block. Then, mice were tested in a randomized training session. Once mice received more than 60 WR in a 1-h session and the average rate of correct choice was above 70% in the randomized training session, mice proceeded to the next phase.

Phase 3: In addition to the target texture (P120 or P400), P180 and P280 were presented as nontarget stimuli in this phase. For example, P120 versus P180, P120 versus P280, and P120 versus P400 were presented when P120 was a target texture. Once mice received more than 60 WR in a 1-h session and the average rate of correct choice was above 70% for three consecutive sessions, mice proceeded to a test session.

## In vitro electrophysiology

Brain slices were prepared and visualized, and whole-cell recordings were performed as described previously [47]. Briefly, three weeks after the virus injection, mice were anesthetized with isoflurane (5% isoflurane (vol/vol) in 100% oxygen); perfused transcardially with an ice-cold sucrose solution containing (in mM) 75 sucrose, 87 NaCl, 2.5 KCl, 26 NaHCO$_3$, 1.25 NaH$_2$PO$_4$, 10 glucose, 0.5 CaCl$_2$, and 2 MgSO$_4$; saturated with 95% O$_2$ and 5%CO$_2$; and decapitated. The brain was rapidly removed from the skull in a bath of ice-cold sucrose solution. Coronal slices of 300 μm were made using a vibratome (Leica Biosystems) and stored in the same solution at 35˚C for 30 min and at 20 to 24˚C for an additional 30 to 45 min before recording.

Whole-cell patch clamp recordings were performed in oxygenated artificial cerebrospinal fluid (ACSF) containing (in mM) 125 NaCl, 2.5 KCl, 26 NaHCO$_3$, 1.25 NaH$_2$PO$_4$, 10 glucose, 2 CaCl$_2$, and 1 MgCl$_2$. The ACSF was equilibrated with 95% O$_2$ and 5% CO$_2$ throughout the entire recording session, which typically lasted between 30 min to 1 h. Recordings were performed at 30 to 33°C. Electrodes (5 to 7 MΩ) were pulled from borosilicate glass capillary (1.5 mm OD). The pipette intracellular solution contained (in mM) 130 potassium gluconate, 6.3 KCl, 0.5 EGTA, 10 HEPES, 5 sodium phosphocreatine, 4 Mg-ATP, 0.3 Na-GTP, and 0.3% neurobiotin (pH 7.4 with KOH, 280 to 290 mOsm). Membrane potentials were not corrected for the liquid junction potential. During patching, cell-attached seal resistances were >1 GΩ. Once whole-cell configuration was achieved, uncompensated series resistance was usually 5 to 30 MΩ, and only cells with stable series resistance (<20% change throughout the recording) were used for analysis. Data were collected using a Multiclamp 700B amplifier (Molecular Devices), low-pass filtered at 5 kHz and digitally sampled at 20 kHz and analyzed with pClamp10 (Molecular Devices). Depolarizing current steps (800 ms duration) were injected to POm neurons under current clamp configuration. Neurons were held at their resting membrane potential (RMP) for all recordings. Action potentials (APs) were quantified and compared before and after bath application (10 min) of CNO (10 μM). RMP, AP frequency, and rheobase were analyzed offline using Clampfit 11.

## Chemogenetic and optogenetic manipulation

**Chemogenetic suppression of POm.**  Saline or CNO (3 mg/kg, i.p.) was injected 30 min prior to the behavioral test. The behavioral task was same as Phase 3. Each saline and CNO session was repeated three times, and two conditions were interleaved.

**Optogenetic suppression of axon terminals from S1, M1/M2, and Sp5i to POm.**  Optical fiber-implanted experimental and control groups were connected to 590 nm LED (power approximately 10 mW at fiber tip). Each LED off and LED on session was repeated three times, and two conditions were interleaved. Light stimulation was triggered by nose poke to a trial initiation port and terminated by nose poke to a water port.

**Combination of chemogenetic and optogenetic suppression.**  Mice were injected with saline or CNO (3 mg/kg, i.p.) 30 min prior to the behavioral task. For the session with saline injection, mice were connected to the LED but the light was off during the entire session. For the session with CNO injection, mice were connected to the LED, and the light was on during each trial.

Whisker trimming: Under light anesthetized condition (1.0% to 1.2% isoflurane, mixed with 1 L per min of oxygen), all the whiskers of both sides were trimmed at least 2 h prior to the behavioral task.

## Animal movement tracking and behavioral data analysis

The movement of the mice was recorded by Plexon infrared video system and analyzed by DeepLabCut [42] with customized deep learning-based behavioral analysis (Python). By labeling a small number of frames (approximately 200 frames/mouse), we trained tailored and robust feature detectors that could localize a variety of experimentally relevant body parts, including the nose, left ear, right ear, center of back, and tail. We then defined the animal's position (initiation, sensory, and two WR zones) and computed movement velocity, time spent in sensory zone, total trial numbers in each session, trial duration, and intertrial interval with custom-written MATLAB scripts.

## Optrode recordings and analysis

In vivo electrophysiological recordings from Rbp4-cre and wild-type mice were performed using a Plexon OmniPlex system. Viral injection (AAV5-EF1a-DIO-eNpHR3.0-eYFP and

AAV5-hSyn-eNpHR 3.0-eYFP) was made into M1/M2 as described above. Prior to recordings, headposts were implanted and the animals were habituated to the head-fixed condition. On the recording day, a small craniotomy and durotomy were made. An optrode (A1x16-5mm-25-177-OA16LP, optic fiber 100 μm, NeuroNexus) was slowly advanced to the recording depth ranging from 2.9 to 3.65 mm from the pia. Electrophysiological signals were acquired through an A32-OM32 adaptor (Neuronexus) and digitized at 40 kHz with a Plexon OmniPlex system. Single units were semimanually sorted offline with Offline Sorter (Plexon). A silicon optrode was coupled to a 590 nm Ce:YAG light source (Doric Lens) via a fiber-optic patch cord (4 mW at the recording tip). Light stimulation was delivered with 2-s duration, 8-s interval (50 to 100 repeats) and controlled by TTL (Pulser, Prizmatix). Data were analyzed and plotted using NeuroExplorer and custom-written MATLAB scripts.

We tested each sorted single unit for a significant decrease in firing rate (bin, 10 ms) during the period of optogenetic stimulation (2 s) compared to a period (2 s) preceding the light stimulation (Wilcoxon rank sum test, one-sided).

### Histology

At the end of the behavioral experiments, all mice were deeply anesthetized and perfused transcranially with 4% paraformaldehyde (PFA) in 0.1 M PBS (pH 7.3). Brains were postfixed in 4% PFA for 2 h at 4˚C and stored in 30% sucrose solutions in PBS at 4˚C. Coronal brain sections (50 μm) were washed in PBS (2 times for 15 min) and incubated in a blocking solution (10% normal goat serum, 1% BSA, and 0.2% Triton X-100 in PBS) for 1 h at room temperature (RT) under constant shaking. Sections were then incubated with primary antibodies cocktail: chicken anti-GFP (Thermo Fisher Scientific, A10262) and rabbit anti-RFP (Rockland, 600-901-379) antibodies in diluted blocking solution (1:10 dilution in PBS) overnight at 4˚C. Sections were rinsed in the diluted blocking solution (3 times for 15 min) before incubated at RT for 2 h in secondary antibodies cocktail. Alexa Fluor 488–conjugated goat anti-chicken (A-11039) and Alexa Fluor 546–conjugated goat anti-rabbit (A-11035) secondary antibodies (1:200 dilution, Thermo Fisher Scientific) were used to visualize fluorescence signals. Sections were rinsed in PBS, mounted on slides, and coverslipped with antifade Fluoromount-G mounting medium containing DAPI (Thermo Fisher Scientific). Fluorescence images were taken by Axio1 Scanner (Zeiss) and A1R confocal microscope (Nikon). After histological verification, mice with either incorrect virus injection or probe implantation were excluded from data analysis.

### Statistics

All average values were indicated as mean ± SEM, and the significance was indicated as * $P < 0.05$, ** $P < 0.01$, *** $P < 0.001$, or **** $P < 0.0001$, unless otherwise noted. Animal number and session number were reported in each figure. Both male and female mice were used, and animals were randomly assigned to each group. Behavioral data were subjected to either independent *t* test (between groups) or paired *t* test (different conditions within individual mouse) using GraphPad Prism and custom written MATLAB scripts. No statistical methods were used to predetermine sample sizes. Data collection and analysis were not performed blind to the experimental conditions.

### Supporting information

**S1 Fig. Timeline of behavioral training and experimental procedure.** Schematic shows the experimental design, including behavioral task training, surgical procedure, behavioral task with optogenetic or chemogenetic manipulation, whisker trimming, perfusion, and

histological evaluation. The different phases of task training are described in the Methods.
(TIF)

**S2 Fig. CNO application significantly suppresses hM4D(Gi)-expressing POm neurons. (A)**
Rheobases of hM4D(Gi)-expressing POm neurons are significantly increased upon CNO
application ($P = 0.002$). Whole-cell patch clamp recording ex vivo was performed on hM4D
(Gi)-expressing POm neurons. Gray bars indicate averaged Rheobase across POm neurons,
and black lines indicate an individual POm neuron. **(B)** Resting membrane potentials of
hM4D(Gi)-expressing POm neurons are significantly decreased upon CNO application
($P = 0.0059$). Plotting conventions are same as (A). **(C)** CNO application decreases evoked
spiking activity from hM4D(Gi)-expressing POm neurons in response to current injection.
The data in S2A-S2C Fig can be found in S1 Data.
(TIF)

**S3 Fig. Expression of hM4D(Gi)-mCherry in POm. (A)** A representative histological image
showing the hM4D(Gi)-mCherry expression in POm. The orange line delineates mCherry
expressing area. Scale bar = 500 μm. **(B)** hM4D(Gi)-mCherry expressing areas are overlayed
for each group. Each different color of shading represents an individual mouse.
(TIF)

**S4 Fig. Chemogenetic suppression of LP does not affect whisker-dependent sensory per-
ception in texture discrimination task.** **(A)** Schematic of experimental design and a represen-
tative image showing the expression of hM4D(Gi)-mCherry in LP. Scale bar = 500 μm. **(B)**
Chemogenetic suppression of LP does not affect task performance ($P = 0.2434$, $n = 6$, paired $t$
test). The data in S4B Fig can be found in S1 Data.
(TIF)

**S5 Fig. Optogenetic suppression of M1/M2 axon terminals in POm.** **(A)** Schematic of opto-
genetic stimulation and extracellular in vivo recording in POm from awake, head-fixed mice.
**(B)** Representative image showing a recording site. Arrowhead indicates an optrode track.
Scale bar = 500 μm. **(C)** Raster plots and peristimulus time histogram (PSTH) of two example
units recorded from POm. 10 ms bin, 60 trials. Shaded areas indicate light stimulation (590
nm, 2 s, 4mW). Illumination with 590 nm significantly decreases the spontaneous spiking
activity. The inset shows overlayed spiking waveforms. Asterisk indicates rebound activity
upon the termination of light stimulation. **(D)** Mean spontaneous activity (z-scored) profile of
POm units in response to optogenetic stimulation of M1/M2 terminals in POm (21 units).
Optogenetic application significantly suppressed spontaneous spiking activity of 21 out of 109
POm units (4 recording sessions from 3 mice). Black line indicates mean, and gray area indi-
cates 95% confidence interval. **(E)** Heatmap of spontaneous activity of individual POm units.
Top orange line indicates optogenetic stimulation. The data in S5C-S5E Fig can be found in S1
Data.
(TIF)

**S6 Fig. Projections from M1/M2 to POm.** **(A)** Schematic of experimental design. AAVretro-
CAG-cre was injected to POm of Ai14 mouse line to label the neurons projecting to POm. **(B)**
Representative images showing tdTomato-positive neurons in M1/M2. Top image represents
a close-up of the L5/6 of M1/M2. **(C)** The laminar distribution of tdTomato-positive cells
within M1/M2 (L5: 251 cells/mm$^2$, L6: 157 cells/mm$^2$, $P < 0.005$, brain tissues $n = 6$, $n = 1$
mouse). **(D)** Schematic of experimental design. Constitutive AAV-hSyn-eNpHR-eYFP was
injected to wild-type mice to unbiasedly target L5/6 cells in M1/M2. **(E)** Representative images
showing the expression of eNpHR3.0-eYFP in M1/M2 (top) and the M1/M2 axon terminals

innervating POm (bottom). (**F**) Optogenetic suppression of M1/M2 axon terminals in POm does not affect task performance ($P = 0.43$, $n = 4$, paired $t$ test). Scale bar = 500 μm. The data in S6 Fig can be found in S1 Data.
(TIF)

**S7 Fig. Suppression of POm activity and S1 axon terminals in POm do not affect behavior parameters of self-initiated texture discrimination task.** (**A**) Mice perform a similar number of trials per session under chemogenetic suppression of POm (left), optogenetic suppression of S1 axon terminals in POm (middle), and combined suppression (right). Gray bars indicate average across mice, and black lines indicate an individual mouse. (**B-D**) Chemogenetic suppression of POm (upper) and optogenetic suppression of S1 axon terminals in POm (lower) do not affect trial duration (**B**), time spent in sensory zone (**C**), and the intertrial interval (**D**). Plotting conventions are same as (**A**). The data in S7 Fig can be found in S1 Data.
(TIF)

**S1 Data. Data underlying figures and Supporting information figures.**
(XLSX)

# Acknowledgments

We thank all members of the Lee lab, the McBain lab, the Petros lab, and the Lu lab for helpful discussions. We thank W. Zhang for behavioral training. We thank the Section on Instrumentation of NIMH (G. Dold, T. Talbot, and D. Ide) and the Center for Information Technology at NIH (J. Krynitsky). We thank the NIMH Systems Neuroscience Imaging Resource for assistance with histological analyses and the NIMH Rodent Behavioral Core for providing surgical setup.

# Author Contributions

**Conceptualization:** Jia Qi, Soohyun Lee.

**Data curation:** Jia Qi, Soohyun Lee.

**Formal analysis:** Jia Qi, Soohyun Lee.

**Funding acquisition:** Soohyun Lee.

**Investigation:** Jia Qi, Changquan Ye, Shovan Naskar, Ana R. Inácio, Soohyun Lee.

**Project administration:** Jia Qi, Soohyun Lee.

**Resources:** Soohyun Lee.

**Supervision:** Jia Qi, Soohyun Lee.

**Validation:** Jia Qi.

**Visualization:** Jia Qi.

**Writing – original draft:** Jia Qi, Soohyun Lee.

**Writing – review & editing:** Changquan Ye, Shovan Naskar, Ana R. Inácio.

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
