## [Editor Report · Decision Letter 0]

5 Jan 2022

Dear Dr Lee, 

Thank you for submitting your manuscript entitled "The role of a higher-order thalamic nucleus in perceptual discrimination" for consideration as a Short Report by PLOS Biology.

Your manuscript has now been evaluated by the PLOS Biology editorial staff, as well as by an academic editor with relevant expertise, and I am writing to let you know that we would like to send your submission out for external peer review.

**IMPORTANT: We note that you have excluded six potential reviewers. The editorial team would like to request that you shorten the list to three (3), you so paper has good chances of being reviewed by the most relevant experts in the field. While we cannot necessarily honor six exclusions, we will honor three. 

In addition, before we can send your manuscript to reviewers, we need you to complete your submission by providing the metadata that is required for full assessment. To this end, please login to Editorial Manager where you will find the paper in the 'Submissions Needing Revisions' folder on your homepage. Please click 'Revise Submission' from the Action Links and complete all additional questions in the submission questionnaire.

Once your full submission is complete, your paper will undergo a series of checks in preparation for peer review. Once your manuscript has passed the checks it will be sent out for review. To provide the metadata for your submission, please Login to Editorial Manager (https://www.editorialmanager.com/pbiology) within two working days, i.e. by Jan 07 2022 11:59PM.

If your manuscript has been previously reviewed at another journal, PLOS Biology is willing to work with those reviews in order to avoid re-starting the process. Submission of the previous reviews is entirely optional and our ability to use them effectively will depend on the willingness of the previous journal to confirm the content of the reports and share the reviewer identities. Please note that we reserve the right to invite additional reviewers if we consider that additional/independent reviewers are needed, although we aim to avoid this as far as possible. In our experience, working with previous reviews does save time. 

If you would like to send previous reviewer reports to us, please email me at ggasque@plos.org to let me know, including the name of the previous journal and the manuscript ID the study was given, as well as attaching a point-by-point response to reviewers that details how you have or plan to address the reviewers' concerns. 

Given the disruptions resulting from the ongoing COVID-19 pandemic, please expect some delays in the editorial process. We apologise in advance for any inconvenience caused and will do our best to minimize impact as far as possible.

Kind regards,

Gabriel

Gabriel Gasque

Senior Editor

PLOS Biology

ggasque@plos.org

---

## [Decision Letter · Decision Letter 1]

2 Mar 2022

Dear Soohyun,

Thank you for submitting your manuscript "The role of a higher-order thalamic nucleus in perceptual discrimination" for consideration as a Short Report at PLOS Biology. Your manuscript has been evaluated by the PLOS Biology editors, by an Academic Editor with relevant expertise, and by three independent reviewers. Please accept my apologies for the delay in communicating the decision below to you.

In light of the reviews (below), we will not be able to accept the current version of the manuscript, but we would welcome re-submission of a much-revised version that takes into account the reviewers' comments. We cannot make any decision about publication until we have seen the revised manuscript and your response to the reviewers' comments. Your revised manuscript is also likely to be sent for further evaluation by the reviewers.

We expect to receive your revised manuscript within 3 months. 

**IMPORTANT - SUBMITTING YOUR REVISION**

Your revisions should address the specific points made by each reviewer. Together with the academic editor, we think that demonstrating the effectiveness of the three different manipulations would require including new data.

Please submit the following files along with your revised manuscript:

*Re-submission Checklist*

*Published Peer Review*

*PLOS Data Policy*

*Blot and Gel Data Policy*

Sincerely,

Gabriel

Gabriel Gasque

Senior Editor

PLOS Biology

ggasque@plos.org

REVIEWS:

Reviewer #1: The paper by Qi et al is timely and important. The authors examined the relevance of the posteromedial (POm) nucleus for tactile perception in mice. Indeed, the roles of POm in tactile perception are not yet clear, and are subject to a significant debate in the field. On the one hand, POm neurons receive massive projections from the whiskers (via brainstem nuclei), from S1, and from M1 (directly as well as via a tunable disinhibition circuit of the ZI). On the other hand, their activity depends strongly on the state of the animal (e.g., anesthetized versus awake head-fixed). Thus, the current study is indeed required.

The strategy of Qi et al. for studying the relevance of POm to tactile perception was to train freely moving mice on a 2AFC texture discrimination task and to see how suppressing POm activity affects performance. After showing that the performance of the mice strongly depends on their whiskers, they showed that it also depends on unperturbed operation of POm neurons. Bilateral suppression of POm significantly impaired performance. This study shows that the default execution of the task involves the POm, as bilateral POm activity suppression was induced only during task execution, and suppression sessions were interleaved with control sessions. The anatomical spread of the manipulation was controlled using labeling in each mouse. While this is probably not the best way to control for the suppression of neurons that are close to the POm-VPM border, it seems good enough - even if only partial suppression of the POm was achieved here, the conclusion still holds. Importantly, in this respect, the authors indeed showed that VPM neurons were not infected (lines 133-136).

Then the authors turned to test the dependency of task performance on 3 major inputs to the POm - S1, M1 and Sp5i. The authors conclude that only the S1->POm projection plays a role in this task, because suppressing the other 2 inputs did not affect task performance. Here I find several significant flaws in the design of the experiment, which call for modifying the conclusive statements accordingly. Otherwise, the authors must verify their conclusions with appropriate controls. 

Major comments

1. Suppression of input from Sp5i. The conclusion of the authors, as stated, requires convincing evidence that this pathway was indeed sufficiently suppressed during the experiment. However, there are significant doubts that this was the case, mostly because: (1) the relevant (rostral) sector of Sp5i (see, for example, El-Boustani et al, 2020; Ebert et al, 2021) might not be sufficiently infected by the virus; (2) the Sp5i terminals in POm may not be sufficiently affected by the laser illumination, which was delivered from fibers located dorsal to the POm. Thus, control experiments, in which the efficiency of suppressing Sp5i terminals is demonstrated, are required here.

2. Suppression of input from M1/M2. Also here, the conclusion of the authors requires convincing evidence that this pathway was indeed suppressed during the experiment. The major potential problems here are: (1) Rbp4-cre may not express Cre in all, or even the most relevant, POm projecting neurons in the motor cortex. Specifically, it is known that corticothalamic neurons in layer L6 of motor cortex project to the POm (Deschenes et al.1998, Harris & Shepherd, 2015,Muñoz-Castañeda et al, 2021); (2) the M1/M2, or ZI, terminals in POm may not be sufficiently affected by the laser illumination spreading from the dorsally located fibers. Thus, control experiments, in which the efficiency of suppressing M1/M2/ZI terminals is demonstrated, are required here.

3. The title should not use the term "higher-order." Rather, simply use "posteromedial" as this is the nucleus that was studied here. There is nothing in the paper that refers generally to "higher-order" nuclei. This distinction is important specifically because there is a debate regarding the "order" of processing among thalamic nuclei - there is substantial empirical evidence that is not consistent with POm being a "higher-order" nucleus (e.g., see Ahissar & Oram 2015). The abstract, and the rest of the paper, can of course refer to the hypothesis that the POm is a "higher-order" nucleus, but as a hypothesis and not as a fact. Referring to it as a fact a) has no empirical justification, b) creates an unjustified bias among the readers, and c) may impair appropriate future empirical designs. 

Minor comments

1. Lines 48-50. "However, the postulated role of POm in encoding whisking behavior has been challenged with studies demonstrating that POm activity is only weakly modulated by whisker movements [3, 27-30]" - not accurate. The two Yu et al studies cited here (2006, 2015) showed robust responses to whisking, though weaker than those measured in the VPM (see also Groh et al 2013). The authors may want to distinguish here between artificial whisking in anesthetized rats and self whisking in head-fixed animals.

2. Fig. 3. Please show the labeling in both hemispheres in the same animals - after all, this is a crucial condition in these experiments.

3. A reference to the stereotaxic atlas used in this study should be added. 

4. Specify to which point the DV coordinates refer? Bregma seems to be distanced from Sp5i by more than "3.6-3.9 mm".

Reviewer #2, Laszlo Acsady: The paper by Qi and colleagues addresses an old puzzle in somatosensation i.e. the role of the higher order or paralemniscal thalamic nucleus, (nucleus posteriomedial or Pom) in sensory discrimination. This question has not yet been resolved despite decades of effort put into clarifying the connectivity and synaptic physiology of Pom. Qi et al use a chemogenetic approach to demonstrate that Pom is involved in sensory discrimination during active sensation and utilize optogenetic suppression to identify the role of its excitatory inputs.

The results are overall highly suggestive. Devil is in the details though. While the suppression of correct responses is significant during Pom blockade the performance of the animals is still well above chance level (70%). This can result from:

A) incomplete inhibition of thalamic neurons by CNO, 

B) not inhibiting the entire Pom

C) small role of Pom in sensory discrimination.

Version A) can be tested by injecting VPM and test the CNO effect. In this case chance level performance is expected. Or by in vivo recording of unit activity.

To exclude B) a much better localization of virus injection site is needed. In order to properly assess the site of CNO action in Pom high resolution, preferably confocal images of the injection sites are needed where it can clearly be determined to what extent Pom was injected. The main problem is that most of the whisker sensitive cells in Pom are located along the VMP/Pom border. This is where S1 layer 5 and Sp5i inputs converge on Pom cells (see e.g Veinante et al. 2000, Lavallee et al., 2005 and Groh et al., 2014 about the convergence). This is even visible on the otherwise very poor quality image of Fig 3E, where the S1 L5 input is clearly more lateral towards the edge of the VPM than the mCherry injection.) The same sector is spared by the injection site on the right side in Fig 2A. this sector is spared by the virus. In addition we don't know the extent of virus injection at different antero-posterior levels. 

Concerning option C). If the authors can positively refute A) and B) the authors should appropriately discuss the relatively small effect of Pom and indicate that according to the data sensory discrimination during active sensation may not be the only role of Pom. 

inputs

Minor points:

It is unclear why "error rate increase" is used in the last chapter (from line 260 on) and not the rate of correct responses. How it is exactly calculated?

Optic fiber tracts appear tobe artificially made in Fig 3.

Reviewer #3: Qi et al present a punchy account of their investigation into the functional role (if any) of the higher-order somatosensory thalamic nucleus POm in sensory perception. This is an important topic that has been argued over for many years. As the authors correctly note, a probable cause for the lack of resolution has to do with the fact that POm has been studied almost entirely under anesthetized or head-fixed conditions. Here the authors add significant clarity to the field by developing a novel two-alternative forced-choice texture discrimination task for head-fixed mice; this in itself is a noteable contribution to the literature. They then use chemogenetic inhibition of POm to probe for a role in the task and find that performance is reduced. This is an important and, as the authors note, surprising finding. One might think that the animals could perform the task at normal levels entirely with the VPM-based pathway, but this is evidently not so.

Next, the authors used optogenetic inhibition of three different inputs to POm, together with or without chemogenetic inhibition of POm, to assay which if any of these inputs is critical for POm's role in task performance. They find that inputs from S1 L5 are critical in that inhibiting these axons recapitulates and occludes the effect of POm silencing. Testing for occlusion in these experiments is an informative and rigorous aspect of the experiments. In contrast to the effect of S1 axonal inhibition, the authors find no effect of a similar inhibition of the ascending inputs from brainstem to POm, nor of descending inputs from M1/M2 to POm. Finally, the authors show that the deficits associated with POm silencing are more pronounced for more difficult texture discriminations, consistent with a role of POm in processing requiring more attention or vigilance.

Overall, the study seems carefully and rigorously designed and executed. One notable exception to the high overall level of rigor has to do with a lack of positive controls for axonal inhibition. Specifically, given the negative results of the manipulations intended to inhibit axons from SpVi -> POm and from M1/M2 -> POm, how do the authors know that they have in fact obtained inhibition of these axons? It would make a much stronger paper if the authors could somehow provide reassurance on this point. At a minimum, they should acknowledge and discuss it.

The authors should also discuss potential caveats associated with their use of halorhodopsin to inhibit axons, which can have unintended effects (B. Mahn, M., Prigge, M., Ron, S., Levy, R., & Yizhar, O. (2016). Biophysical constraints of optogenetic inhibition at presynaptic terminals. Nature neuroscience, 19(4), 554-556. https://doi.org/10.1038/nn.4266).

The writing and overall presentation is quite clear. A couple minor suggestions:

-It would help the reader if the authors showed a schematic of the task sequence in Figure 1A. It is described in the text, but a visual aid would add clarity. 

-Section heading titled "The role of POm in sensory perception". I would suggest changing this to "A role of POm in sensory perception". The authors present evidence that POm plays some role, but their data don't have much to say about what exactly this role is.

---

## [Decision Letter · Decision Letter 2]

21 Sep 2022

Dear Dr Lee,

Thank you for your patience while we considered your revised manuscript "The role of a posteromedial thalamic nucleus in perceptual discrimination" for consideration as a Short Reports at PLOS Biology. Your revised study has now been evaluated by the PLOS Biology editors, the Academic Editor and 2 of the original reviewers. 

In light of the reviews, which you will find at the end of this email, we are pleased to offer you the opportunity to address the remaining comments from Reviewer 1 (Ehud Ahissar) in a revision that we anticipate should not take you very long, specifically doing some rewriting to clearly discuss the suggestions that Reviewer 1 raises, explicitly note the strengths and limitations that the current data set has for supporting/refuting the hypothesized functions of the POm and providing any additional data you might have in hand to address the lingering issues. We will then assess your revised manuscript and your response to the reviewers' comments with our Academic Editor aiming to avoid further rounds of peer-review.

While revising this submission, please also make sure to address all of the data and other policy-related requests found at the bottom of this email under the Resubmission Checklist heading. At this time, we'd also suggest you consider a slight modification to the title of your piece. We'd suggest:

"Posteromedial thalamic nucleus activity significantly contributes to perceptual discrimination."

**IMPORTANT - SUBMITTING YOUR REVISION**

*Resubmission Checklist*

*PUBLISHED PEER REVIEW*

*ETHICS STATEMENT*

Your study requires an ethics statement - and we note that one is already included in the methods section where you've written: “All procedures were in accordance with protocols approved by the NIMH/NIH Animal Care and Use Committee.”

DATA POLICY:

Note that we do not require all raw data. Rather, we ask that all individual quantitative observations that underlie the data summarized in the figures and results of your paper be made available. The goal is to allow readers to re-run the analysis for an individual figure based on data the authors used. 

These can be supplied in one of the following forms:

Fig1C-E; Fig2B,D; Fig3B,D,F-H; Fig4A-C

Supplemental: Fig2A-C; Fig4B; Fig5C-E; Fig6C-F; Fig7A,B

**Please ensure that your Data Statement in the submission system accurately describes where your data can be found.

For an example see here: http://www.plosbiology.org/article/info%3Adoi%2F10.1371%2Fjournal.pbio.1001908#s5

THIS STEP IS OFTEN FORGOTTEN!

*BLURB*

Please provide a blurb which, if the paper is accepted, will be included in our weekly and monthly Electronic Table of Contents (eTOCs), sent out to readers of PLOS Biology. This blurb may also be used to promote your article on social media. The blurb should be about 30-40 words long and is subject to editorial changes. It should, without exaggeration, entice people to read your manuscript, should not be redundant with the title and should not contain acronyms or abbreviations. For examples, view our author guidelines: https://journals.plos.org/plosbiology/s/revising-your-manuscript#loc-blurb

*PROTOCOLS DEPOSITION*

Sincerely,

Kris

Kris Dickson, Ph.D. (she/her)

Neurosciences Senior Editor/Section Manager

PLOS Biology

kdickson@plos.org

REVIEWS:

Reviewer's Responses to Questions

Do you want your identity to be public for this peer review?

Reviewer #1: Yes: Ehud Ahissar

Reviewer #2: Yes: László Acsády PhD

Reviewer #1: The authors should be lauded for their extensive efforts to address the reviewers' concerns.

I think that the results are much more convincing now. And thus, it is important to be accurate in drawing conclusions from these results, and specifically conclusions about the possible functions of POm in tactile perception.

As far as my original comments are concerned, we talk basically on the following issue. The authors show that while the integrity of POm is important for performing the task, the integrity of its inputs from M1/M2 and from Sp5i is not.

Now let's examine what is the meaning of this result, given existing hypotheses for the functions of the lemniscal and paralemniscal systems.

Regarding the lemniscal system (VPM and S1 in this case) the accepted hypothesis is that it is the major system for processing the spatial features of external objects. This hypothesis is nicely confirmed here.

Regarding the paralemniscal system there are 2 major hypotheses: I. it is the major system for processing/controlling whisker motion during tactile perception, II. It is not conveying any peripheral information that is relevant to tactile perception (and instead the POm is used as a cortico-cortical bridge). Let's see what the current results mean for these 2 hypotheses.

Apparently, the results seem to support hypothesis II, and they are interpreted in this way by the authors. However, careful examination shows that the results cannot discriminate between the two hypotheses.

According to Hypothesis I, lesions of the inputs to the POm should affect the control of whisking, but not texture discrimination. The only way in which such lesions could significantly affect texture discrimination is if they block all relative motion between the whiskers and the object. It could be that performance would also be impaired (slightly, or temporarily) if there is still motion after the lesion but with a modified pattern (modified whisking speed, frequency, amplitude, head-whisker relationships, etc.). Had the authors measured these parameters their comparative analysis could tell us if this was the case and if so, how it affects performance.

Thus, based on their data, the authors are in no position to rule out the possible functions of POm in "sensory perception", as they do. POm can still play a role in sensory perception as suggested by Hypothesis I above. The authors should make this fact clear throughout the article, anywhere where such interpretations are described (I found them in several places). This is crucial in order not to mislead future experimental designs in this field.

Specific points related to same issue:

1. With a 2AFC sensory discrimination task, and performance level ~80%, there are many possibilities for the mice to adapt to motor or motor-related losses. Thus, the mice could easily, and very rapidly, change their motor strategy (all is required is some relative motion between the whiskers and the objects) in order to maintain reasonable performance. Thus, in general in my opinion, a study aimed at addressing vibrissal perception must include careful, high-resolution tracking of whiskers and head kinematics, and their palpation patterns over the textures.

2. I also think that generalizing from one simple 2AFC sensory discrimination task to "sensory perception" is unacceptable. Statements should refer to the specific task used.

Reviewer #2: The authors adequately answered all my concerns. Congratulations for the revision. Nice work.

---

## [Editor Report · Decision Letter 3]

28 Oct 2022

Dear Dr Lee,

Thank you for the submission of your revised Short Reports "Posteromedial thalamic nucleus activity significantly contributes to perceptual discrimination" for publication in PLOS Biology. On behalf of my colleagues and the Academic Editor, Mathew Diamond, I am pleased to say that we can in principle accept your manuscript for publication, provided you address any remaining formatting and reporting issues. These will be detailed in an email you should receive within 2-3 business days from our colleagues in the journal operations team; no action is required from you until then. Please note that we will not be able to formally accept your manuscript and schedule it for publication until you have completed any requested changes.

When submitting your final revisions, please also make the following changes. These are required before we can formally accept your submission:

1) Please ensure that each of the figure legends **in your manuscript** include a statement indicating where the underlying data for that figure can be found (e.g. "The data in Fig 1, panel A can be found in the excel file...")

2) Please correct the ethics statement on the Details page of our website. I see that you currently say "not required". Your study in mice does require such a statement, and one is included in your methods section already. So we just need you to update this notice on the website.

Please also take a minute to log into Editorial Manager at http://www.editorialmanager.com/pbiology/, click the "Update My Information" link at the top of the page, and update your user information to ensure an efficient production process.

PRESS

We frequently collaborate with press offices. If your institution or institutions have a press office, please notify them about your upcoming paper at this point, to enable them to help maximize its impact. If the press office is planning to promote your findings, we would be grateful if they could coordinate with biologypress@plos.org. If you have previously opted in to the early version process, we ask that you notify us immediately of any press plans so that we may opt out on your behalf.

Sincerely, 

Kris Dickson, Ph.D., (she/her)

Neurosciences Senior Editor/Section Manager

PLOS Biology

kdickson@plos.org